# Optimización en dos pasos del rendimiento del software usando transformaciones de compilación

**Juan Carlos de la Torre**
Escuela Superior de Ingeniería
Universidad de Cádiz
juan.detorre@uca.es

**José Miguel Aragón-Jurado**
Escuela Superior de Ingeniería
Universidad de Cádiz
josemiguel.aragon@uca.es

**Javier Jareño**
Escuela Superior de Ingeniería
Universidad de Cádiz
javier.jareno@uca.es

**Bernabé Dorronsoro**
Escuela Superior de Ingeniería
Universidad de Cádiz
bernabe.dorronsoro@uca.es

**Patricia Ruiz**
Escuela Superior de Ingeniería
Universidad de Cádiz
patricia.ruiz@uca.es

## Abstract

Los compiladores tradicionales permiten generar versiones más eficientes de un programa mediante el uso de *flags* de optimización, que aplican secuencias específicas de transformaciones de código. Sin embargo, estas secuencias suelen ser genéricas y a menudo no logran su objetivo de mejorar el rendimiento. El impacto de las transformaciones aplicadas, así como su orden, depende tanto del *software* como del *hardware* en el que se ejecuta. Este hecho hace necesario la generación de secuencias de transformaciones *ad hoc* para la optimización del rendimiento del *software*. En este trabajo definimos un nuevo problema compuesto por dos problemas de optimización combinatoria distintos, cuyo objetivo es encontrar secuencias que optimicen el rendimiento del programa, y que tengan la menor longitud posible. Reducir la longitud de la secuencia óptima permite realizar compilaciones más rápidas utilizando únicamente las transformaciones de código más beneficiosas para el binomio *hardware-software*, generando así nuevo conocimiento sobre este importante tema. Se propone una metodología de dos pasos para resolver este problema. En el primero, se identifican secuencias óptimas que minimizan el tiempo de ejecución del programa objetivo. En el segundo, se minimiza la longitud de la secuencia tratando de mejorar aún más el rendimiento del programa. Los resultados muestran cómo el método propuesto encuentra secuencias hasta un 51,27% más cortas y con una mejora del rendimiento en tiempo de ejecución en más de un 26% en comparación con el *flag* de compilación -O3.

## 1 Introducción

El rendimiento eficiente del *software* es esencial para maximizar el uso de los recursos disponibles en las plataformas *hardware*, un factor clave para lograr prácticas de computación más sostenibles. Los compiladores tradicionales aplican secuencias genéricas de transformaciones de código para optimizar atributos como el tiempo de ejecución, el uso de memoria o el consumo de energía. Sin embargo, estos enfoques genéricos suelen ser ineficaces [1], ya que no tienen en cuenta la gran

XVI XVI Congreso Español de Metaheurísticas, Algoritmos Evolutivos y Bioinspirados (maeb 2025).

diversidad y rápida evolución del *hardware*, cada uno con características y requisitos únicos [2–4]. Esta ineficiencia resalta la necesidad de metodologías innovadoras que aprovechen las características específicas del programa y los recursos computacionales disponibles para optimizar su rendimiento.

El tipo de transformaciones realizadas sobre el código, así como su orden, afectan significativamente al rendimiento del *software*, ya que dependen de la interacción entre el *software* y el *hardware*. La infraestructura de compilación LLVM [5] con sus 87 transformaciones disponibles (versión 9.0.1), también llamadas *passes*, proporciona herramientas para optimizar el *software* a nivel de representación intermedia (*Intermediate Representation* - IR) manteniendo su funcionalidad. Sin embargo, identificar la secuencia óptima de estas transformaciones y el orden en que deben aplicarse representa un reto combinatorio [6]. Además, secuencias de transformaciones más cortas no solo mejoran la eficiencia de la compilación, sino que también facilitan el descubrimiento de optimizaciones eficaces.

Predecir el impacto de las transformaciones de código en el rendimiento del programa es realmente complejo [7]. Aunque estas transformaciones suelen tener un bajo impacto directo sobre el rendimiento del *software*, su influencia puede volverse significativa cuando se combinan con otras. En [7] se muestra cómo el efecto directo de las transformaciones es generalmente inferior al 2%, pero llega a alcanzar el 70% al combinarse con otras. Esta disparidad entre el efecto directo y el combinado de las transformaciones resalta la complejidad del problema. Además, se conoce que el impacto de las transformaciones varía sustancialmente entre diferentes programas y arquitecturas [8].

En este trabajo, abordamos el desafío planteado mediante la aplicación de dos tareas de optimización de manera secuencial. La primera consiste en encontrar secuencias de transformaciones que reduzcan el tiempo de ejecución, conocido como *Software Code Optimization Problem - SCOP* [8]. La segunda tarea busca reducir la longitud de la mejor secuencia encontrada sin empeorar el rendimiento, que denominamos *Sequence Length Minimization Problem - SLMP*. Este enfoque permite mejorar la eficiencia computacional y descubrir las transformaciones más relevantes.

Este trabajo presenta varias contribuciones. En primer lugar, se define un nuevo problema de optimización combinatoria a dos niveles para encontrar las secuencias de transformaciones de código más cortas que minimicen el tiempo de ejecución del *software*. En segundo lugar, se propone una metodología de dos pasos para resolver el problema, que consiste en (i) la identificación de la secuencia de transformaciones que minimiza el tiempo de ejecución de un programa objetivo en una plataforma *hardware* específica y (ii) la reducción de la longitud de la secuencia sin empeorar su rendimiento. En tercer lugar, se lleva a cabo una validación experimental en una arquitectura x86 utilizando dos programas distintos, demostrando la efectividad práctica del enfoque propuesto. En cuarto lugar, se realiza un análisis sobre las transformaciones utilizadas en las secuencias obtenidas.

En cuanto a la estructura del documento, la Sección 2 presenta los principales trabajos relacionados con nuestro estudio. La Sección 3 define el nuevo problema de optimización propuesto. La Sección 4 describe la metodología seguida en este trabajo, mientras que la Sección 5 detalla la configuración experimental. Finalmente, la Sección 6 presenta los resultados obtenidos, y la Sección 7 expone las conclusiones y posibles líneas de trabajo futuro.

## 2  Estado del arte

Reducir el tiempo de ejecución del *software* implica realizar un uso más eficiente de los recursos disponibles, como pueden ser la CPU, la memoria RAM y los periféricos [9]. Las optimizaciones a nivel de compilador son especialmente valiosas para reducir los tiempos de ejecución. Estas optimizaciones pueden eliminar operaciones redundantes de memoria o reorganizar bucles para mejorar el uso de los registros, incrementando tanto el rendimiento como la eficiencia energética [10].

La optimización del *software* puede abordarse como un problema de tipo *phase ordering problem*, que consiste en determinar la secuencia óptima en la que deben aplicarse las transformaciones de código de un compilador para mejorar el rendimiento del *software*. Debido a la compleja interacción entre las transformaciones, encontrar una secuencia óptima supone un desafío *NP-hard* [6, 11]. Mientras que las primeras investigaciones se centraron en minimizar el tamaño del código para ahorrar almacenamiento y memoria [6], trabajos posteriores se han enfocado en la reducción del tiempo de ejecución [11] y en explorar formas de mitigar los elevados costes de compilación que conlleva este problema [12].

Por otro lado, la incertidumbre o ruido es un factor crítico en diversos campos, como la visión artificial y los sistemas embebidos, y no abordarla adecuadamente puede provocar fallos graves [13]. En los sistemas embebidos resulta esencial encontrar un equilibrio entre robustez y eficiencia [14]. Según [15], la incertidumbre puede afectar a las variables de decisión, a la función de *fitness* o al entorno en sí. Para gestionar estos desafíos, se han desarrollado enfoques que van desde métodos robustos basados en intervalos [16, 17] hasta modelado metaheurístico y gestión de ruido en la función de *fitness* [18], que ayudan a reducir errores en las estimaciones de rendimiento. En este sentido, [19] presenta un estudio exhaustivo sobre estas estrategias. Cabe destacar que el modelado basado en intervalos suele requerir un gran número de muestras, lo que lo hace computacionalmente costoso. Para mitigar este problema, a veces se utiliza bootstrapping para generar conjuntos de muestras amplios con un coste computacional reducido [8].

Como las transformaciones de código ofrecen un impacto limitado en el rendimiento cuando se aplican de forma aislada, los niveles estándar de optimización de los compiladores (por ejemplo, -O1, -O2, -O3) agrupan alrededor de 300 transformaciones para obtener mejoras significativas [7]. Sin embargo, una secuencia más corta y cuidadosamente seleccionada puede influir considerablemente en el tiempo de ejecución. En este estudio se trabaja en esta dirección, considerando la incertidumbre presente en el tiempo de ejecución durante los procesos de optimización.

## 3   Definición del problema

En esta sección se introduce un nuevo problema de optimización en dos niveles, compuesto por dos problemas combinatorios que deben resolverse de forma secuencial. El primer problema, denominado SCOP [8], consiste en identificar la secuencia de transformaciones de código genéricas que minimizan el tiempo de ejecución de un *software* específico en un determinado *hardware*. El segundo problema se centra en refinar la secuencia óptima de transformaciones obtenida en el primer paso, tratando de reducir su tamaño sin empeorar el rendimiento del programa.

Formalmente, para el primer paso, dado un programa $P$ y un conjunto de transformaciones de código genéricas $T = \{t_1, \ldots, t_k\}$, cada una de las cuales puede transformar $P$ en un código semánticamente equivalente, el problema se define como encontrar la secuencia de transformaciones $\vec{S} = [s_1, \ldots, s_n] \in T$ (donde $n$ puede ser mayor, menor o igual a $k$) de manera que el tiempo de ejecución del código resultante $P'$ sea minimizado:

$$\text{Minimize } SCOP(P, \vec{S}) = Runtime(P'), \tag{1}$$

donde $P'$ es el resultado de aplicar todas las transformaciones en $\vec{S}$ a $P$, en el mismo orden en el que aparecen en $\vec{S}$, y donde $s_i \in T$. La misma transformación puede aplicarse varias veces ($s_i = s_j, i \neq j$), ya que suele provocar cambios adicionales, tal y como se establece en [7, 8, 20].

Posteriormente, para el segundo problema de optimización, que denominamos *Sequence Length Minimization Problem* (*SLMP)*, se resuelve un problema de selección de transformaciones. Partiendo del programa $P$ y de la secuencia de transformaciones $\vec{S}$ obtenida en el paso anterior, el objetivo es determinar una secuencia binaria $\vec{B} = [b_1, \ldots, b_n]$ que selecciona qué transformaciones de $\vec{S}$ deben aplicarse para obtener una subsecuencia $\vec{S'}$ que minimice el tiempo de ejecución de $P$. Formalmente:

$$\text{Minimize } SLMP(P, \vec{S}, \vec{B}) = SCOP(P, \vec{S'}), \tag{2}$$

donde $\vec{S'}$ es una subsecuencia de $\vec{S}$, compuesta por las transformaciones indicadas en $\vec{B}$, que genera una versión semánticamente equivalente del programa $P$ con el tiempo de ejecución mínimo.

En la Figura 1 se presenta una visión general del problema propuesto. Como se muestra, el primer paso de optimización (SCOP) identifica la secuencia óptima de transformaciones de código seleccionadas de entre todas las opciones disponibles (algunas pueden repetirse), con el objetivo de minimizar el tiempo de ejecución de un *software* en su arquitectura *hardware* objetivo. A continuación, el segundo paso (SLMP) busca la subsecuencia óptima sin empeorar el tiempo de ejecución.

## 4   Metodología

A continuación, se describe la metodología seguida en este trabajo. La Sección 4.1 ofrece una visión general de la infraestructura de compilación LLVM, que proporciona transformaciones genéricas del

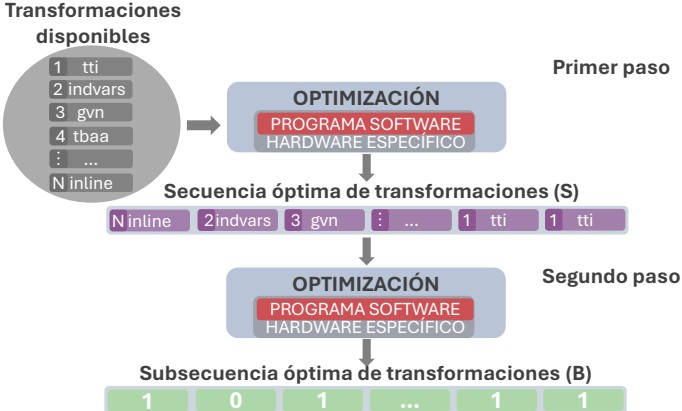

Figura 1: Ilustración del proceso en dos pasos para optimizar el tiempo de ejecución de un *software* específico en una plataforma *hardware* determinada minimizando la longitud de la secuencia.

código fuente. Posteriormente, la Sección 4.2 introduce el algoritmo de optimización empleado para abordar el nuevo problema propuesto.

## 4.1 Infraestructura de compilación LLVM

LLVM es una infraestructura de compilación que cuenta con un lenguaje similar al ensamblador, llamado Representación Intermedia (*Intermediate Representation* - IR), que se genera a partir del código fuente mediante el compilador Clang. LLVM ofrece numerosas operaciones de análisis y optimización de código, conocidas como *passes*, que preservan su semántica. Estos *passes* se aplican sobre el código IR, por lo que pueden utilizarse para optimizar *software* en los distintos lenguajes y arquitecturas *hardware* soportados por LLVM. Por tanto, el uso de transformaciones de LLVM en el código IR garantiza que la metodología sea genérica y aplicable a cualquier *software* desarrollado en lenguajes compatibles (por ejemplo, C, C++, Rust, etc.).

El optimizador de LLVM proporciona tres tipos *passes*: i) *análisis* (recopilan información sobre el *software* para depuración o para otros *passes*), ii) *transformación* (modifican el programa manteniendo su funcionalidad), y iii) *utilidad* (proporcionan información adicional de ayuda). En este trabajo se utilizan los de transformación, que modifican el código fuente y pueden aplicarse múltiples veces. Además, se tienen en cuenta los *passes* de análisis necesarios para la aplicación de algunos *passes* de transformación.

El comando *opt*, responsable de aplicar las transformaciones, actúa como principal optimizador y analizador de LLVM. Inicialmente, el compilador Clang genera el código IR de LLVM a partir del código fuente original. A continuación, se aplica una secuencia de *passes* al código IR para modificarlo, utilizando *opt*. Finalmente, Clang se usa nuevamente para compilar el código IR optimizado en un ejecutable para el *hardware* objetivo, asegurando que no se introduzcan modificaciones adicionales.

## 4.2 Algoritmo genético celular

En este trabajo, se propone el uso de un algoritmo genético celular (Cellular Genetic Algorithm - cGA) [21] para resolver de manera eficiente el nuevo problema planteado. En comparación con otros GA tradicionales, el cGA es capaz de mantener la diversidad de la población durante más tiempo [22], mitigando así la convergencia prematura. Los cGAs se caracterizan por su población estructurada, donde los individuos están organizados en una malla toroidal, lo que introduce el concepto de distancia entre individuos dentro de la población.

El pseudocódigo del cGA utilizado en este trabajo se presenta en el Algoritmo 1. El proceso comienza generando una población de individuos aleatoria, que luego son evaluados para determinar sus valores de *fitness* (Línea 2). Tras esta inicialización, el algoritmo entra en un bucle evolutivo (Líneas 3 a 15), donde cada iteración representa una generación. En cada generación, se aplican operadores evolutivos a los individuos de la población uno a uno (Líneas 6 a 12). Se identifican los vecinos del individuo

---
**Algorithm 1** Pseudocódigo de un cGA canónico
---
1: **proc** cGA()
2: pob = nuevaPoblacion();
3: **while not** condicionDeParada() **do**
4:     **for** x = 1 **hasta** ANCHO **do**
5:         **for** y = 1 **hasta** ALTO **do**
6:             n_list = computaVecinos(pob,x,y);
7:             padre1 = Seleccion(n_list);
8:             padre2 = Seleccion(n_list);
9:             descendencia = Recombinacion(padre1, padre2);
10:            Mutacion(descendencia);
11:            Evaluacion(descendencia);
12:            Reemplazo(pob,x,y,descendencia);
13:         **end for**
14:     **end for**
15: **end while**
16: **end proc** cGA;
---

actual (Línea 6) y se seleccionan dos padres de la vecindad, asegurando que sean distintos (Líneas 7 y 8). Estos padres se someten a un cruce con cierta probabilidad para generar un descendiente (Línea 9). A continuación, se aplica el operador de mutación con una probabilidad baja (Línea 10). Se evalúa la solución resultante (Línea 11) y, finalmente, sustituye al individuo actual en la población según una política de reemplazo definida (Línea 12).

## 5 Configuración de los experimentos

Además de las transformaciones disponibles en LLVM, se introduce una transformación adicional, que denominamos NONE. Esta transformación no modifica el código IR, y su inclusión permite que el algoritmo pueda encontrar secuencias de transformaciones más cortas que la longitud de cromosoma predefinida en el primer problema de optimización. Este algoritmo se ejecuta 30 veces para cada experimento con el fin de obtener resultados estadísticamente confiables. Se ha adoptado la misma configuración de parámetros a la propuesta en [8] para resolver SCOP con un cGA, detallada en la Tabla 1.

Table 1: Parámetros de configuración para la experimentación

| Parámetro | SCOP | SLMP |
|---|---|---|
| Longitud de los individuos | 300 | Longitud de la secuencia (ls) |
| Tamaño de la población | 100 | 100 |
| Forma de la población | $10 \times 10$ | $10 \times 10$ |
| Inicialización de la población | Aleatoria | Aleatoria |
| Vecindad | L5 | L5 |
| Selección de padres | Individuo + Torneo Binario | Individuo + Torneo Binario |
| Cruzamiento (*Recombinación*) | Cruce de dos puntos | Cruce de dos puntos |
| Probabilidad de cruzamiento | $\rho_c = 1.0$ | $\rho_c = 1.0$ |
| Mutación | Valor aleatorio | Inversión de bit |
| Probabilidad de mutación | $\rho_0 = 1/300$ | $\rho_0 = 1/ls$ |
| Reemplazo | Reemplazo si mejora | Reemplazo si mejora |
| Ejecuciones independientes | 30 | 30 |

Para la evaluación del *fitness*, se emplea el método wcase15 [8], que ha demostrado ser muy eficaz para gestionar la incertidumbre en las mediciones del tiempo de ejecución durante la optimización.

Para evaluar la metodología, se emplean dos conjuntos de pruebas (*benchmarks*) distintos. Uno es *Polybench* [23], utilizado en [8], que consiste en la ejecución secuencial de un subconjunto de programas de *Polybench*. Incluye una amplia variedad de operaciones, excluyendo tareas extremadamente costosas computacionalmente para evitar tiempos de experimentación excesivos. El otro *benchmark*

es *ConsumerEmbench* [20], compuesto por la ejecución simultánea de una selección de programas de Embench [24], diseñados para sistemas embebidos de bajo consumo.

El algoritmo de optimización ha sido implementado en Python 3 utilizando la biblioteca jMetalPy [25]. Todos los experimentos se han realizado en un sistema *hardware* con un procesador Intel Core i5-8210Y (con 2 núcleos físicos a 1.6–3.6 GHz) y 8GB de RAM. El sistema operativo utilizado es Ubuntu 22.04 y se ha usado la biblioteca estándar de GNU C (glibc) versión 2.35.

## 6 Resultados

En esta sección se presentan los resultados obtenidos en la resolución del nuevo problema en cada uno de los *benchmarks*. En la **Sección 6.1**, se evalúa la calidad de las soluciones encontradas en términos de tiempo de ejecución y longitud de la secuencia de transformaciones. Posteriormente, en la **Sección 6.2**, se analiza la utilidad de las transformaciones más utilizadas y de las menos utilizadas en las mejores secuencias obtenidas por el cGA en cada uno de los 30 experimentos realizados. Por último, analizamos el efecto del paso de reducción del tamaño de la secuencia sobre la mejor secuencia encontrada en nuestros experimentos.

### 6.1 Evaluación de la calidad de las soluciones encontradas

La Figura 2 muestra la evolución de la población del cGA en la optimización de SLMP sobre la mejor secuencia encontrada al resolver SCOP, para ambos conjuntos de pruebas. Como se puede observar, hay una mejora progresiva en la calidad de la solución a lo largo de las 100 generaciones. Además, en la Figura 2(a), se observa que en *Polybench* los niveles de diversidad siguen siendo altos al final de la evolución, lo que indica que el valor de *fitness* aún podría mejorar con ejecuciones más largas.

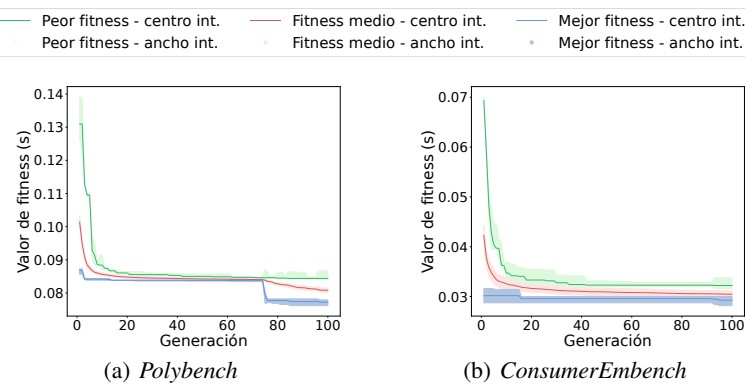

Figura 2: Evolución de la mejor solución encontrada al resolver SLMP a lo largo de 100 generaciones.

Para comparar la calidad de las soluciones obtenidas, ejecutamos $1.000$ veces la mejor versión del *software* encontrada en las 30 ejecuciones independientes del cGA. El mismo proceso se repite para cada *benchmark* compilado con los *flags* de LLVM `-O0` (sin optimización) y `-O3`. Los resultados obtenidos se presentan en la Figura 3. En ambos casos, la mejor solución obtenida supera claramente tanto al programa no optimizado como al compilado con el *flag* `-O3` con diferencias estadísticamente significativas. Para *Polybench*, la solución obtenida mejora el tiempo de ejecución en un $46.42\%$ respecto al programa no optimizado y en un $10,79\%$ en comparación con la optimización `-O3`, reduciendo además el número de transformaciones en un $48,73\%$. Para *ConsumerEmbench*, la solución mejora el tiempo de ejecución en un $76.56\%$ respecto al programa sin optimizar y en un $51,14\%$ frente a `-O3`, reduciendo el número de transformaciones en un $51,27\%$.

También nos interesa analizar el impacto de la reducción de la secuencia de transformaciones en el rendimiento del programa. La Figura 4 compara el rendimiento de la mejor solución encontrada, antes y después del paso SLMP para ambos *benchmarks*, realizándose $1.000$ ejecuciones independientes de cada uno. En ambos casos, la reducción de la secuencia provoca una pérdida de rendimiento del $12.24\%$ en *Polybench* y del $4.34\%$ en *ConsumerEmbench*. Sin embargo, a la hora de valorar estas diferencias es conveniente tener en cuenta la alta incertidumbre presente y los cortos tiempos de

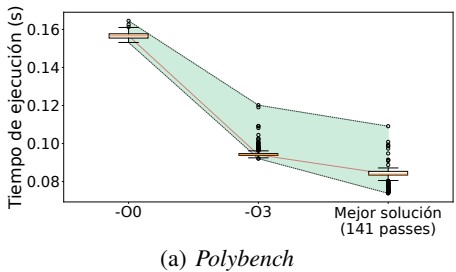

(a) *Polybench*

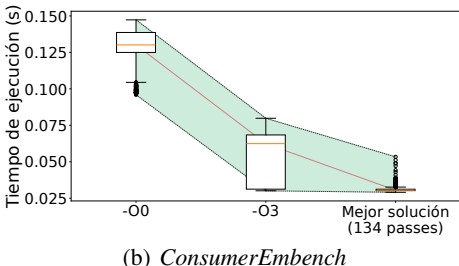

(b) *ConsumerEmbench*

Figura 3: Comparación del rendimiento del mejor resultado obtenido por el cGA (sobre 1.000 ejecuciones) frente al mismo *software* compilado con `-O0` (es decir, sin optimizaciones) y `-O3`.

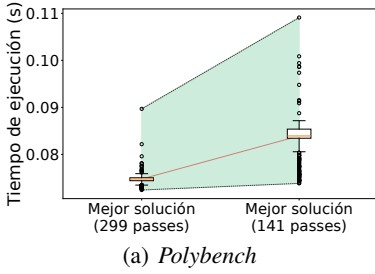

(a) *Polybench*

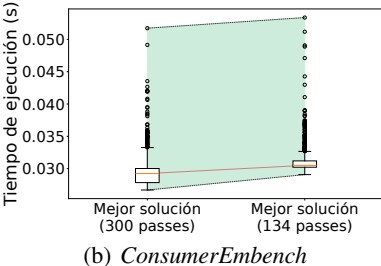

(b) *ConsumerEmbench*

Figura 4: Comparación del rendimiento del mejor resultado obtenido por el cGA (sobre 1.000 ejecuciones) frente a la solución de SCOP, antes de la minimización de la secuencia (SLMP).

ejecución. Por ejemplo, para *Polybench*, donde las diferencias son mayores, se observa que un gran número de valores atípicos en la solución reducida se sitúan cerca del límite inferior de rendimiento de la solución no reducida. Además, la secuencia de transformaciones se reduce el 52, 84% en *Polybench* y el 55, 33% en *ConsumerEmbench*.

## 6.2 Análisis de la utilidad de las transformaciones

Esta sección presenta un estudio sobre la utilidad de las transformaciones encontradas en las mejores secuencias obtenidas. En primer lugar, se analiza el número de veces que cada transformación es aplicada en las soluciones generadas. La Figura 5 ilustra el número promedio de apariciones de las diez transformaciones más utilizadas en cada uno de los dos *benchmarks* estudiados, para las 30 ejecuciones del cGA. Se observa que cada transformación se emplea más de dos veces en promedio. Además, ambos *benchmarks* solo comparten tres transformaciones entre las diez más utilizadas: `-loop-rotate` (reestructura los bucles para mejorar el rendimiento), `-jump-threading` (elimina flujos de control innecesarios), y `-licm` (mueve código invariante de bucles fuera de ellos).

En el caso de las 10 transformaciones menos utilizadas en cada *benchmark*, hemos observado que ninguna transformación se aplica más de 1, 4 veces en promedio. Para *ConsumerEmbench*, todas las transformaciones disponibles se aplican al menos una vez en promedio, lo que resalta la complejidad de encontrar una secuencia óptima de reducido tamaño. En cambio, en *Polybench*, las transformaciones `-instcombine` (combina múltiples instrucciones para simplificarlas) y `-gvn` (elimina cálculos redundantes) no se aplican en ninguna de las soluciones obtenidas en las 30 ejecuciones del cGA. Además, las soluciones obtenidas para ambos *benchmarks* comparten solo dos transformaciones entre las menos utilizadas: `-memdep` (que realiza análisis de dependencias de memoria), y `-elim-avail-extern` (que elimina funciones externas innecesarias del programa).

Finalmente, la Figura 6(a) ilustra el número de veces que se aplica cada transformación en la mejor solución encontrada para SCOP sobre *Polybench*, así como tras su optimización con SLMP. Se observa que el algoritmo descarta transformaciones de todos los tipos al reducir la solución. Durante el proceso de SLMP, el algoritmo tiende a priorizar la reducción de las transformaciones más repetidas, equilibrando finalmente el número de aplicaciones entre todos los tipos. Esto pone de

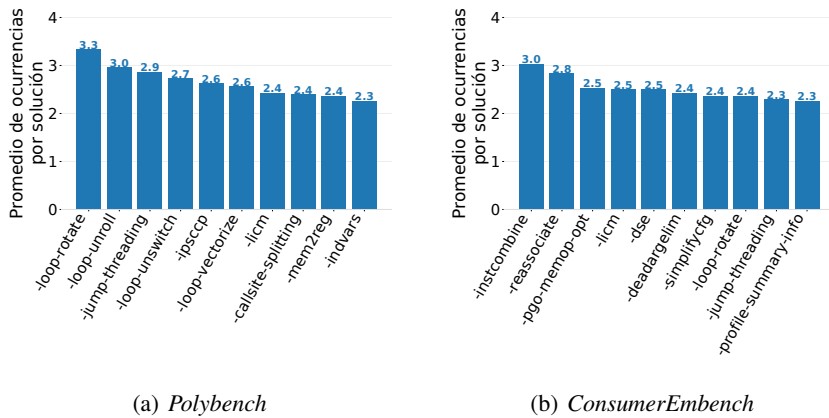

(a) *Polybench*  (b) *ConsumerEmbench*

Figura 5: Los 10 *passes* más utilizados en las soluciones obtenidas en las 30 ejecuciones del cGA.

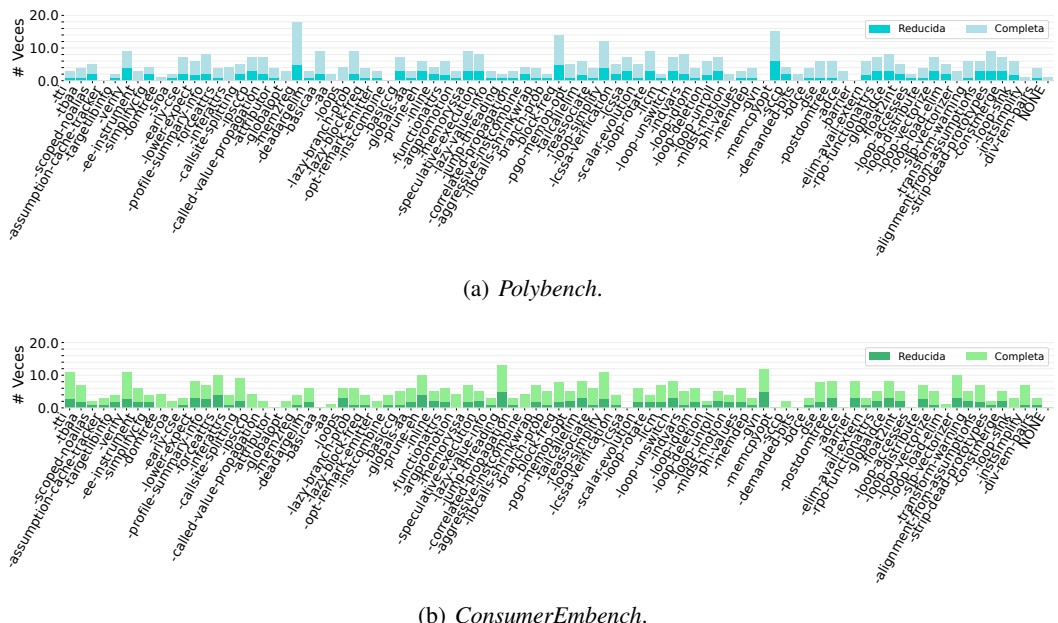

(a) *Polybench*.

(b) *ConsumerEmbench*.

Figura 6: Frecuencia de aplicación de cada transformación en las mejores soluciones encontradas, antes y después del proceso de reducción del tamaño de la secuencia (SLMP).

manifiesto la dificultad de identificar qué transformaciones tienen un mayor impacto en el rendimiento del programa. En total, diez *passes* fueron eliminados completamente de la solución obtenida en *Polybench*, incluyendo: `-ee-instrument`, `-domtree`, `-aa`, `-loops` y `-demanded-bits`, que realizan análisis de código; `-inferattrs` y `-globalopt`, que optimizan atributos y variables; `-mldst-motion`, que reorganiza accesos a memoria; y `-slp-vectorizer` y `-instsimplify`, que optimizan simplificaciones y vectorización de instrucciones.

En cuanto a la mejor solución para *ConsumerEmbench*, la Figura 6(b) muestra un comportamiento similar durante el segundo paso de optimización, donde el número de transformaciones aplicadas para cada tipo se equilibra. En este caso, un total de 12 transformaciones han sido completamente eliminadas en la solución reducida: `-domtree`, `-aa` y `-opt-remark-emitter` (centradas en el análisis del código); `-sroa`, `-ipsccp`, `-called-value-propagation`, `-argpromotion` y `-sccp` (dedicadas a la promoción y simplificación de valores); `-globalopt` y `-aggressive-instcombine` (aplican optimizaciones globales al programa); y `-loop-load-elim` y `-loop-sink` (realizan opti-

mizaciones específicas de bucles). En ambos *benchmarks* se eliminan los *passes* `-domtree`, `-aa` y `-globalopt` (principalmente enfocados en el análisis del código y la optimización de variables).

En resumen, el algoritmo de minimización tiende a equilibrar el número de transformaciones aplicadas de cada tipo en los *benchmarks* estudiados, reduciendo en mayor medida aquellas que se repiten con más frecuencia. Finalmente, las marcadas diferencias entre las transformaciones más y menos utilizadas en cada benchmark ponen de manifiesto la complejidad del problema abordado, ya que el impacto de las transformaciones depende en gran medida de las características del *software*, lo que resalta la necesidad de optimizaciones *ad hoc* en el proceso de compilación del *software*.

## 7 Conclusiones

En este trabajo, presentamos un nuevo problema en dos etapas centrado en identificar secuencias mínimas de transformaciones de código para optimizar el rendimiento en tiempo de ejecución del *software*. Este problema implica dos tareas de optimización secuenciales. Primero, se determina la secuencia óptima de transformaciones que minimiza el tiempo de ejecución del programa mediante la resolución de SCOP. Luego, esta secuencia se refina mediante un problema de optimización, denominado SLMP, con el objetivo de mantener o incluso mejorar el tiempo de ejecución del programa, al tiempo que se acelera el proceso de compilación.

Abordamos este nuevo problema aplicando un algoritmo genético celular a cada una de las tareas de optimización, y la incertidumbre se gestiona mediante la técnica *wcase15* de la literatura, que emplea *bootstrapping* y una evaluación de *fitness* basada en intervalos. El enfoque propuesto se prueba en dos *benchmarks* distintos ejecutados en un dispositivo con arquitectura x86.

Las soluciones obtenidas logran una mejora de hasta un 76.56% en el tiempo de ejecución respecto al programa no optimizado y hasta un 51.14% en comparación con la secuencia genérica de optimización `-O3` de LLVM, además de reducir el número de transformaciones en un 51.27%. De los experimentos realizados se desprende que si bien se puede reducir la secuencia de transformaciones para optimizar el programa en alrededor de la mitad, esto acarrea un cierto impacto negativo en el rendimiento, de entre el 4% y el 12%, aproximadamente, en los estudios realizados. Estos datos deben valorarse a la hora de plantear un problema que resuelva simultáneamente SCOP y SLMP. En cuanto a las secuencias de transformaciones obtenidas, el proceso de reducción tiende a equilibrar el número de transformaciones de cada tipo utilizadas en la solución, priorizando la eliminación de aquellas aplicadas con mayor frecuencia. Además, se observaron diferencias significativas entre las transformaciones más y menos utilizadas en cada *benchmark*. Estos resultados subrayan la necesidad de una nueva generación de compiladores inteligentes capaces de diseñar secuencias de optimización *ad hoc* más cortas. Nuestra principal línea de trabajo futuro se centra en el diseño de nuevos algoritmos para la optimización simultánea de los dos problemas considerados. Asimismo, se plantea ampliar el estudio utilizando programas de *software* más complejos y otros dispositivos de hardware.

### Agradecimientos

Esta publicación forma parte del proyecto PID2022-137858OB-I00, financiado por MI-CIU/AEI/10.13039/501100011033 y por "FEDER Una manera de hacer Europa", y del proyecto eFracWare (TED2021-131880B-I00), financiado por MICIN/AEI/10.13039/501100011033 y la Unión Europea NextGeneration EU/PRTR. J. M. Aragón-Jurado agradece el apoyo del Ministerio de Ciencia, Innovación y Universidades de España a través del contrato FPU21/02026.

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
