# OpenReview forum: "Optimización en dos pasos del rendimiento del software usando transformaciones de compilación"
_MAEB/2025/Congreso — MAEB 2025_

### Official Review · Reviewer_n4yK · 2025-03-13
**Optimización en dos pasos del rendimiento del software usando transformaciones de compilación.**

**Rating:** 3
**Confidence:** 4

**Review:**

En este trabajo los autores proponen un nuevo problema de optimización, encontrando secuencias que optimicen el rendimiento de un programa minimizando la longitud de estas secuencias. Para resolverlo, se propone una metodología en dos fases. En la primera se minimiza el tiempo de ejecución del programa y en la segunda se minimiza la longitud de la secuencia sin afectar al rendimiento.

Cambios mayores:

- Uno de los autores ha hecho evidente su autoría en los agradecimientos. Cuando se presenta un trabajo con un proceso de revisión anonimizado debe prestarse especial atención a este tipo de detalles, puesto que está rompiendo con la filosofía del proceso.

- El trabajo está bien escrito en líneas generales, pero debe revisarse en profundidad para asegurar que las frases tienen sentido en castellano. Da la impresión de que el paper estaba parcialmente escrito en inglés y se ha pasado por algún mecanismo de traducción, lo que hace que haya algunas frases que quedan extrañas en castellano (la primera frase del abstract, por ejemplo, no se escribiría así en castellano). Los autores deben re-escribir el paper completo para evitar estas imprecisiones.

- En el abstract, los autores afirman proponer un nuevo problema compuesto por dos problemas de optimización distintos. Más adelante, en la Introducción, afirman que se define un nuevo problema, que se corresponde con el SLMP. Una de las dos afirmaciones es imprecisa: o el nuevo problema es el SLMP o el nuevo problema es la combinación de SLMP y SCOP. En mi opinión, sería más adecuado afirmar que se propone el SLMP y que se resolverán SLMP y SCOP de manera simultánea. Es decir, se están resolviendo dos problemas de optimización, uno propuesto con anterioridad por los propios autores y otro que se propone en este trabajo.

- La solución se prueba en una arquitectura x86. Sin embargo, la tendencia actual de los procesadores apunta hacia ARM64 de manera clara, siendo esta arquitectura la presente en la mayoría de dispositivos móviles, IoT, e incluso en los últimos procesadores de Apple. ¿Se plantea ampliar el estudio a esta arquitectura?¿Funciona la propuesta en estos entornos? Sería interesante explorarlo.

- En la definición del problema, nuevamente se afirma proponer un nuevo problema de optimización compuesto por dos problemas. No estoy de acuerdo con esta afirmación, el problema SCOP se propone previamente en otro trabajo, y en este problema se propone el SLMP. Sin embargo, en la definición del problema se afirma de manera explícita que resolver el SLMP equivale a resolver el SCOP en una subsecuencia. En mi opinión, si esta definición es cierta, no se está proponiendo entonces un nuevo problema, sino que se están resolviendo dos problemas de optimización diferentes, y cabe la duda de si se están realmente resolviendo dos problemas diferentes o dos problemas equivalentes, con lo cual, en definitiva, se está resolviendo un problema que ya existía. Los autores deben clarificar muy bien este aspecto, al ser esta una de las contribuciones del trabajo. No se puede considerar esta aportación si realmente se está proponiendo un problema ya propuesto (y, por lo tanto, resolviendo un problema ya resuelto), más aún cuando ese problema fue propuesto por los propios autores. En definitiva: si no cambia la función objetivo ni las restricciones del problema, no se puede considerar que se esté proponiendo un problema nuevo.

- El pseudocódigo del Algoritmo 1 es impreciso. ¿ANCHO y ALTO son parámetros del procedimiento cGA? Hay que cambiar ! por not. El pseudocódigo no debería ser tan cercano al código fuente, sino especificar el comportamiento del algoritmo a más alto nivel.

- En la sección 6.1, línea 197, se afirma que: "Para comparar la calidad de las soluciones obtenidas, la mejor solución de las 30 ejecuciones independientes del cGA para cada conjunto de pruebas se ejecuta un total de 1.000 veces". Esta frase debe ser más clara. ¿Se refiere a que la secuencia obtenida se aplica un total de mil veces? ¿Esto tiene sentido cuando no hay ninguna componente aleatoria en la aplicación de las secuencias? Si los autores no se refieren a esto, ¿lo que se ejecuta 1000 veces es el algoritmo que resuelve el SLMP? Si es así, lo que se ejecuta no son soluciones, sino el algoritmo. La acción de "ejecutar soluciones" me resulta extraña, quizá no esté comprendiendo algo correctamente. Entiendo que lo que se hace es ejecutar el binario compilado después del proceso mil veces, midiendo el tiempo de ejecución y reportando el promedio, pero es una intuición, no algo que extraiga del manuscrito y podría estar equivocado. Recomiendo aclarar este punto.

- En la Figura 2, recomiendo utilizar un color diferente para indicar el ancho int. del peor fitness (la función representada en verde). El color elegido no se aprecia hasta hacer un zoom de 350%.

- Uno de los objetivos de la propuesta es reducir las optimizaciones aplicadas sin afectar al rendimiento del programa resultante. Sin embargo, se observa una pérdida de rendimiento en ambos conjuntos de pruebas. Me surge la duda de si se cumple este objetivo en vista de los resultados arrojados por la experimentación.

- Resulta muy interesante el análisis final respecto a los optimizadores utilizados y el número de veces que se utiliza cada uno. Sugiero ampliar en anchura la figura 6, o al menos ampliar el tamaño de la fuente y los espacios entre ticks del gráfico.

Typos y cambios menores:

- En la página 2, línea 39: La razón en que, --> La razón está en que,
- En la página 3, línea 102: tratando de reducir sun tamaño --> tratando de reducir su tamaño
- Hay que ser consistente con el uso de cursiva. En la línea 174 de la página 5 se escribe Polybench en cursiva, y en la 175 de la misma página sin ella. En la línea 195, página 6, vuelve a utilizarse cursiva. En la misma página, en las líneas 202 y 211 se utiliza de nuevo sin cursiva. Por favor, unificar (también en las etiquetas de las figuras).

---

### Official Review · Reviewer_sJBj · 2025-03-17
**Poca relevancia para los temas de interés del congreso.**

**Rating:** 1
**Confidence:** 4

**Review:**

Este trabajo presenta una nueva formulación para un problema de optimización que tiene como objetivo mejorar el proceso de compilación de sistemas software, resultando en programas con un menor tiempo de ejecución. Se presenta un algoritmo general para resolver el problema propuesto y se comparan los resultados satisfactoriamente con el flag de compilación -O3 del compilador.

Uno de los principales problemas del trabajo es que el foco está en la formulación del problema. No hay novedad en el método propuesto ni en la forma de aplicarlo al problema. El trabajo tiene poca relevancia para los temas de interés del congreso.

El problema de optimización definido extiende un problema de optimización ya estudiado en la literatura. La extensión realizada se podría beneficiar de una mayor justificación. El primer problema, SCOP, se define como "*encontrar la secuencia mínima de transformaciones […] de manera que el tiempo de ejecución […] sea minimizado*". Por lo tanto, el segundo problema está contenido en el primero. Más adelante, el segundo problema, SLMP, se define como "*selecciona[r] qué transformaciones […] deben aplicarse para obtener una subsecuencia […] que minimice el tiempo de ejecución[…].*" A mi entender, no hay diferencia entre el primer problema y el segundo. SCOP es equivalente a SLMP, tal y como está definido.

Sin embargo, asumamos que SCOP trata de minimizar el tiempo de ejecución y que SLMP trata de minimizar el número de pases aplicados sin empeorar la eficiencia. Parece que, en realidad, esto es un problema bi-objetivo, no un problema en dos niveles. Si tenemos en cuenta el detalle de "sin empeorar la eficiencia", se entiende que la eficiencia del programa resultante es el objetivo principal, mientras que la longitud de la secuencia es un objetivo secundario. En otras palabras, la longitud de la secuencia podría ser un criterio de desempate en SCOP. Si dos secuencias dan como resultado dos programas con la misma eficiencia, entonces la que tiene menor número de pases es la mejor secuencia.

Parece que la definición de un problema en dos niveles es una formulación innecesaria para este contexto, y que de hecho puede llevar a peores resultados, ya que la minimización de la longitud de la secuencia de pases parte de una secuencia ya dada que se ha encontrado considerando únicamente la eficiencia, limitando el espacio de búsqueda del segundo problema. Otra cosa sería que el método propuesto tuviese dos fases diferenciadas en las que primero busca soluciones considerando la eficiencia y luego intenta minimizar la longitud de las secuencias. Pero esto sería parte del diseño del método, no de la formulación del problema. Creo que (i) una aproximación bi-objetivo o (ii) la introducción del número de pases como criterio de desempate en el primer problema serían aproximaciones más correctas y más sencillas. ¿Por qué han decidido los autores formular un problema en dos niveles?

El objetivo del segundo problema es minimizar el número de pases porque esto dará lugar a menores tiempos de compilación, tal y como se dice en el abstract. ¿Por qué no tener en cuenta directamente el tiempo de compilación? Si realmente hay beneficios en considerar el número de pases en vez del tiempo de compilación (probablemente por el tiempo que se tarda en evaluar una solución), aun así es necesario analizar la correlación entre número de pases y tiempo de compilación. ¿Existe una correlación positiva entre ambas métricas? Si es así, ¿cómo es la correlación? ¿Cómo escala el tiempo de compilación con respecto al número de pases? ¿Y qué influencia tienen los pases utilizados (más que la longitud de la secuencia) en el tiempo de compilación?

En las líneas 137 a 140, se especifica que solo se utilizan los pases de transformación, que son los que modifican el código fuente. Sin embargo, muchos pases de transformación necesitan que previamente se hayan ejecutado algunos pases de análisis en particular. Más aún, es posible que haya que volver a ejecutar algunos de estos pases de análisis después de algunos pases de transformación, ya que el código cambia. ¿Se ha tenido esto en cuenta?

La mejora del tiempo de ejecución mostrada en la Figura 2 es considerable en términos porcentuales, pero mínima en términos absolutos. ¿Se mantiene esta mejora porcentual en programas de mayor complejidad y tamaño? ¿O la mejora es también de unas pocas décimas en programas con tiempos de ejecución más largos?

En la línea 203, se dice que la solución para Polybench mejora el tiempo de ejecución de este programa en un 86,64% respecto al programa no optimizado. El programa no optimizado tiene un tiempo de ejecución aproximado de 0,16. El programa generado con la solución propuesta tiene un tiempo de ejecución de 0,08. ¿Cómo han calculado los autores esa mejora del 86,64%?

La sección de agradecimientos, por desgracia, elimina la posibilidad de una revisión doble-ciego.

Detalles menores:
- Línea 102: "*reducir **sun** tamaño*".

---

### Official Review · Reviewer_pTKG · 2025-03-17
**Un trabajo interesante con espacio de mejora en presentación**

**Rating:** 5
**Confidence:** 4

**Review:**

El trabajo muestra una aplicación interesante de los algoritmos evolutivos para la optimización de código durante compilación. La metodología experimental parece sólida, gracias al empleo de tests estadísticos. Sería necesario sin embargo dar más información al respecto (que tipo de test se ha empleado, y cuál es el p-valor crítico).

La presentación del trabajo necesita mejorarse. Hay pasajes con construcciones gramaticales que parecen calcos del inglés. La primera frase del abstract es un ejemplo muy claro, pero no es el único. Sería conveniente que los autores releyeran el trabajo e intentaran emplear un lenguaje más natural y fluido.

Comentarios menores:
- línea 39, la razón *es*
- línea 79, la mención a la incertidumbre necesita clarificación. Incertidumbre ¿en qué sentido o sobre qué?
- línea 115, "tiempo de ejecución" -> "tamaño"

---

### Decision · Program_Chairs · 2025-03-20

Accept